# Thermodynamic Sensitivity of Blood Plasma Components in Patients Afflicted with Skin, Breast and Pancreatic Forms of Cancer

**DOI:** 10.3390/cancers14246147

**Published:** 2022-12-13

**Authors:** Andrea Ferencz, Dávid Szatmári, Dénes Lőrinczy

**Affiliations:** 1Department of Surgical Research and Techniques, Heart and Vascular Center, Faculty of Medicine, Semmelweis University, Nagyvárad Square 4, H-1089 Budapest, Hungary; 2Department of Biophysics, Medical School, University of Pécs, Szigeti Street 12, H-7624 Pécs, Hungary

**Keywords:** DSC, deconvolution, breast cancer, melanoma malignum, pancreas tumor

## Abstract

**Simple Summary:**

Based on the World Health Organization’s global survey, conducted over the past five years, cancer is the second leading cause of death. According to its database, breast cancer is the third, melanoma malignum is fifth and pancreatic tumors rank as the twentieth. Undoubtedly, the early diagnosis and monitoring of these tumors and related research is important for both healthcare systems and societies, worldwide. In the present study, we compared the deconvoluted components of these blood plasma differential scanning calorimetry curves in patients with solid tumors. Our measurements showed individual, yet disease-specific, curves obtained from DSC measurements of patients’ blood plasma, which were associated with disease severity, progression, or response to treatment. Further research is necessary to elucidate these results in order to raise the possibility of an early diagnosis of a potential tumor or for testing the efficacy of the therapy from a drop of blood.

**Abstract:**

According to the World Health Organization’s 2018 Global Cancer Survey, cancer is the second leading cause of death. From this survey, the third most common is breast cancer, the fifth is melanoma malignum and pancreatic adenocarcinoma ranks twentieth. Undoubtedly, the early diagnosis and monitoring of these tumors and related research is important for aspects of patient care. The aim of our present review was to explain an impressive methodology that is deemed suitable in reference to studying blood sample deviations in the case of solid tumors. Essentially, we compared the heat denaturation responses of blood plasma components through differential scanning calorimetry (DSC). In the control, between five and seven separable components can be detected, in which the primary component was albumin, while in the case of tumorous patients, the peaks of immunoglobulins were dominant. Moreover, the shape of the plasma DSC curves changed with a shift in the higher temperature ranges; thus, their pattern can be used as a suitable marker of direct immunological responses. The further development of the analysis of DSC curves raises the possibility of the early diagnosis of a potential tumor, the monitoring of diseases, or testing the efficacy of the therapy from a single drop of blood.

## 1. Introduction

### 1.1. Studying Blood Plasma with DSC

Over the past fifty years, the medical/clinical application of simple, inexpensive, easy-to-implement robotic thermal analytical methods has gradually increased. The trend began in 1973 with calorimetric measurements performed on red blood cells (rbc) by Monti and Wadsö. They demonstrated the difference, based on heat production, between normal and anemic blood samples [1] prior to, during, and following the treatment in hyperthyroid patients [2], as well as the effect of methylene stimulation [3] including pH, temperature, glucose concentration and storage conditions [4]. This group also performed a careful comparison between different calorimetric techniques, suspension media and preparation methods [5]. Hernández et al. tested rbc, first in the case of patients afflicted with an advanced non-small cell lung tumor [6].

The technical improvement in differential scanning calorimetry (DSC) ushered in an opportunity to simplify sampling and achieve quicker measurements (automatic sample change). In clinical applications, Monaselidze and his team, in 1997, marked the first attempt regarding blood serum and plasma on patients afflicted with cancer [7,8]. In 2006,Garbett’s Lab and Michnik’s Lab were the first to discover the main target regarding plasma is the albumin [9,10]. Garbett’s team were the first to conduct the analysis of whole plasma proteome, in which they declared DSC plays an important role in the clinical diagnosis and monitoring of different cases [11,12,13,14,15,16,17,18,19,20,21,22,23]. From the point of view of diagnosis, it is important to know which thermodynamic parameter corresponds to the response of different plasma components and how they change across different types of diseases in order to establish an accurate diagnosis. Garbett’s group proved remarkable in this area [24,25,26,27] by developing data processing methods which are understandable and easy to use, even for the practicing physician. This also motivates physicians to implement the DSC method on blood plasma to gain a precise diagnosis. Most of the research groups working to use DSC as a diagnostic tool also apply their statistical evaluation.

The most active participants who adhere to Garbett’s pioneering idea originate in Europe. Garbett’s first followers, in chronological order, were a Polish group [10], who recently drew important physiological conclusions regarding athletes and their coaches from blood plasma following physiotherapy treatments, particularly for the fitness status of those who participate in various sports [28,29,30,31,32]. The second are the Hungarian group of this study. Based on our productive collaboration with clinicians, we have new groups of PhD students enrolled in the clinical area, consistently, every year. Our research profile covers a very wide spectrum of medical/biological application, including: patients afflicted with various tumors [33,34,35,36,37,38,39,40]; patients in different stages of COPD [41,42]; the identification of the different stages of psoriasis and the effect of their drug treatment [43,44,45,46,47], including the follow-up regarding the consequences of chemotherapeutic treatment following a successful tumor operation in blood plasma [48,49]. The upcoming Bulgarian lab is intended to apply DSC to the investigation of multiple myeloma cases [50,51,52,53,54] in patients afflicted with colorectal cancer [55]. Another group from Bulgaria [56,57,58,59] made superb contributions in both neurodegenerative disorders and in lung cancer [60].

### 1.2. Breast Cancer

Today, breast tumors are the leading cause of death among females worldwide with an increasing annual rate and it has a high risk level, predominantly in developed countries [61,62]. Due to highly advanced surgical oncology, the recent rate of survivability is over 90% regarding early-stage diagnosis [63,64]. The diagnosis/prognosis of breast cancer is reliant on conventional techniques common to mammography, CT and MRI; however, following biopsy, the detection of CEA (Carcinoembryonic Tumor Marker) and CA15-3 (Carcinoma Antigen 15-3) or different biomarkers such as Her2 (human epidermal growth factor receptor-2) and CK5 (cytokeratin-5) are required. The DSC analysis of blood plasma obtained from patients with newly diagnosed and operable breast cancer can serve as an important method for determining suitable indicators in reference to the size of tumors and number of local metastatic lymph nodes.

### 1.3. Pancreas Cancer and Pancreatitis

Dysfunctions of the pancreas are considerably diverse and include acute and chronic pancreatitis, including pancreatic cancer. Acute pancreatitis, in most cases, appears as a sudden attack, and as pancreas inflammation is an internal medicine case, surgical intervention is generally not justified. In the chronic status of pancreatitis, we can observe the irreversible destruction of acinar and islet cells, evoking tissue fibrosis, which is often due to alcoholism [65,66]. In this stage, a majority of patients will undergo surgery. The pancreas tumor has one of the highest mortality and lowest survivability rates following surgery. Surgical resection of the tumor is classified into three groups: R0 for resection as a cure or complete remission; R1 for a microscopic residual tumor; and R2 for a macroscopic residual tumor. The R classification considers clinical and pathological findings. A reliable classification requires the pathological verification related to the margins of resection. In the R0 stage, the resectability of the tumor also infers curative operability. If the predictable status is R1 or R2, surgical intervention may only result in palliative therapy. In the prehospitalization of patients, the applied methods are imaging tests (X-ray, CT, MRI and US), labor blood test (e.g., CA 19.9 tumor marker with limited diagnostic value, and liver functions), and taking biopsies pre- or intra-operatively [67]. Recently, we had no diagnostic laboratory test regarding the early state of chronic pancreatitis exacerbation, e.g., a serum amylase and lipase level may be normal or only mildly elevated. Furthermore, it is difficult to perform most of pancreatic function tests as they are not routinely available and are not specific in reference to precise diagnosis [68]. There are no current recommended screening programs in support of the general population. Unfortunately, the early detection of pancreatic cancer is rarely successful. We can recommend a DSC blood plasma test to find a new way to help in this area [38].

### 1.4. Skin Cancer

Worldwide, there is an increasing incidence rate regarding the most malignant tumor of the skin (cutaneous melanoma) [69,70]. The most important risk factors in the development of MM are endogenous (genetic markers, skin type) and exogenous conditions (e.g., UV irradiation) [71]. In terms of survivability, early detection is important. In advanced stages, there is a poor prognosis and low survivability. Early treatment involves the surgical removal of MM and adjuvant therapy (chemo-, immuno- and/or radiation therapy) of the tumor [72]. In reference to the determination of the pathological stage of MM, in 1969, Clark et al. proposed a staging criterion which was based on their tissue invasion level [73]. Later, Breslow substantiated the importance of the primary melanoma thickness in millimeters. The Breslow’s index is one of the most important prognostic indicators in the case of ulceration, mitosis and regression [74]. Recently, Sentinel lymph node biopsy became a compulsory step for patients if the thickness of their tumor is more than 1 mm [75,76]. There is an increasing need for further studies and methods to monitor MM patients.

### 1.5. Application of Blood Plasma DSC in Diagnostics

In addition to general changes to blood plasma, we endeavored to determine which plasma components change due to developing diseases. Our aim is the search for a possible link between the type of cancer and the level of blood plasma components as ideally good target indicators of immunological responses within cancer formation. The deconvolution of the DSC curves showed consequential changes (e.g. Table 1) in the thermal denaturation of tumorous blood plasma. In the implementation of the Sanchez-Ruiz’s method [77], in order to achieve the activation energy (e.g. Table 2), we attempted to describe the stability and denaturation kinetics of protein’s structure. Certainly, the determination of the kinetic parameters from the analysis of curves recorded under similar denaturing conditions are strongly conditioned by the kinetic analysis of the experimental data. Thus, we endeavored to find miniscule differences which can truly describe a process to calculate the kinetic parameters with accuracy. The change of activation energies distinguishes the minor sequence and structural differences of blood plasma proteins with similar denaturation responses, supposing it ushers in new information regarding the different immunological processes due to cancer formation. 

## 2. Materials and Methods

### 2.1. Patient Population

The DSC analysis of blood plasma obtained from patients afflicted with breast, pancreatic and skin cancer can serve as an important method in determining good indicators regarding their severity and stages. The aspects of their selection and classification are mentioned in the relevant sections in the Results and Discussion section.

The protocols for breast cancer studies were approved by the Regional Research Ethical Committee of Clinical Centre at University of Pécs (3601.316-12736/2009) in patients afflicted with pancreas diseases (4425/2012) and for skin cancer (27.06.2008/3220).

### 2.2. Blood Sample Collection and Preparation

Peripheral blood samples were collected from patients and from non-tumorous control individuals. The blood samples were stored and sealed in Vacutainer tubes containing EDTA (1.5 mg mL^−1^ of blood) and were then centrifuged at 1.600× *g* for 15 min at 4 °C to separate the plasma fraction from the cell components. The native plasma samples were stored at −80 °C until the DSC measurements were taken.

### 2.3. DSC Measurements

The thermal unfolding of the human plasma components was monitored by SETARAM Micro DSC-II calorimeter, as previously described [12]. All of the experiments were conducted between 0 and 100 °C. In all cases, the heating rate was 0.3 K min^−1^. Conventional Hastelloy batch vessels were used during the denaturation experiments with a 950 μL sample volume, on average. The reference sample was normal saline (0.9% NaCl). The sample and reference samples were equilibrated with a precision of ±0.1 mg. The repeated scan of the denatured sample was used as baseline reference, which was subtracted from the original DSC curve. We plotted the heat flow (DSC-II is a heat flux instrument with hermetically closed vessels) as a function of temperature. Calorimetric enthalpy was calculated from the area under the heat flow curve by using two-point setting SETARAM peak integration.

### 2.4. Deconvolution of DSC Thermal Curve

The collected DSC scans were deconvolved with Gaussian fitting in the accuracy range of R^2^ = 0.986–0.999, typically set to find 5–7 peaks in the relevant temperature intervals [14,55]. For the deconvolution, we applied Origin 2018 (OriginLab, Northampton, MA, USA). The contribution of a few of the Gaussian components was less than the error of instrument enthalpy determination (<5%) and, thus, were omitted. 

### 2.5. Calculation of Activation Energy

With the exception of low molecular mass globular proteins, denaturation is irreversible in biological systems. The first model of irreversible denaturation kinetics of protein systems is named after Lumry and Eyring [77], which was further developed by Sanchez-Ruiz et al. [78,79,80], as well as by Vogl et al. [81]. We used the model of Sanchez-Ruiz. The collected infinitesimal DSC enthalpy change data (dH, average of three independent measurements) were integrated (dH_cal_) in the function of time related change of the temperature. Next, the ln(ln(dH_cal_/dH_cal_ − dH)) vs. 1/T function was fitted with a linear straight line [79]. The melting temperature (T_m_) was the result of the ratio of the slope and its intercept. The slope multiplied by the Regnault number is equal to the activation energy (E_a_).

## 3. Results of Thermal Analysis

### 3.1. Breast Cancer Studies

In Figure 1, we presented our results which related to the classification by the number of metastatic lymph nodes. Group A: 0 affected lymph node; Group B: 1–3 metastatic lymph nodes (Sentinel lymph node positivity); Group C: 4–10 metastatic lymph nodes. Another aspect of classification is based on the tumor’s diameter (Figure 2). In each group, the size was increased by 10 mm, and the blood plasma DSC scans were analyzed, as previously published [33].

Patients only belonged to two groups. In Group 2 (Gr2), the tumor diameter was between 11 and 20 mm, while in Group 4 (Gr4) it ranged between 31 and 40 mm. Ten healthy controls and 19 females with verified breast carcinoma were involved in this study (median age 55.4 years).

We applied the method of Briere [82] for the determination of the deconvolved components’ contribution to the total calorimetric enthalpy using Figure 1 and Figure 2. Figure 1 represents the DSC scans of patients with or without metastatic lymph nodes, the control samples and tumorous samples depict six thermal transitions (T_m_) following deconvolution. Beyond the absolute number of the transitions, the second peak represents the dominant component in the control sample; however, in the case of the tumorous samples (Groups A–C), the third peak was the main contributor. Samples from tumorous patients in the group of without metastatic lymph nodes (Group A) show the second largest peak with the fourth *T*_m_, while in the case of patients with a large number of lymph node positivity (Group C), it shifted to the fifth transition. 

In Figure 2, the blood plasma DSC scans and their deconvolved components were presented for the control and breast cancer patients’, representative of Gr2 and Gr4. We decomposed the original DSC curves to six components in the control group, and also in Gr2, however, in the case of Gr4, it resulted in only five components. There is a remarkable difference between the shape of the curves from both the control and tumorous samples. In terms of the size of the tumors, the width of the peaks from the higher temperature ranges were increased. 

In the case of the control sample, in particular, the second component, noted in the cancer group as the third transition, were the main contributors. 

As seen in Table 1 and Table 3, (made after [83]), different stages of breast tumor-related blood plasma components [10,14,32,55,82] responded with typical thermal transitions in the ranges of globulin free and fatty acid free albumins, between 50–57 °C, 60–63 °C, and 66–69 °C, which overlaps with the range of immunoglobulins as haptoglobulins, α-2 globulins, γ globulins, in IgG and IgA between 70–72 °C, 73–75 °C, and 76–79 °C. 

To apply another thermodynamic parameter which can indicate the effect of cancer regarding the stability and denaturation kinetics of plasma components is the activation energy provided by the Sanchez-Ruiz’s method [78] (Table 2 and Table 4). Interestingly, the activation energy decreased if a metastatic node existed and increases with their number. However, the activation energy was increased if the tumor size reached the range of 11–20 mm and decreased with the size of the tumors (Table 2 and Table 4).

### 3.2. Pancreas Cancer and Chronic Pancreatitis Studies

In Figure 3, in the case of chronic pancreatitis, the deconvolved DSC curve contained more transitions than the control or tumorous samples. The main component was the third transition (T_m3_), and when compared to the control, it was the second peak (T_m2_). The first two contributors were shifted to the lower temperature range (by 7 °C) compared to the controls. This indicates that there was some structural loss in the plasma constituents. In contrast, the third denaturation temperature was increased by 5 °C, indicating that it became more stable, requiring more energy to initiate the structural change. Presumably, these changes reflect the plasma state in a serious systemic inflammatory process. From the previously identified plasma proteins sampled among healthy individuals, the first (T_m1_) transition is primarily attributable to fibrinogen and T_m2_ to albumin, while Tm_3_ is attributable to immunoglobulins [14]. These observations are consistent with the generally accepted and proven facts which state that albumin levels decrease and fibrinogen levels increase during inflammatory processes. In addition, the calorimetric enthalpy of the plasma was slightly decreased, which may be due to a response to the destabilization of several plasma proteins, likely due to globulins, which may play a primary role in inflammation [38]. Seemingly, it is not simply a difference in their quantity, it is also a change in the way protein interacts with inflammatory processes. This is the first time the DSC method has been used to investigate such changes in the blood plasma of patients afflicted with a chronic inflammatory disease such as chronic pancreatitis. Some investigators have carried out similar DSC measurements in soft tissue inflammation, rheumatoid arthritis, in septic and nonseptic arthritis or in different psoriasis stages, as in other systematic inflammatory diseases [14,15,16,17,18,19,43,44,45,46,55].

In terms of tumorous cases, the shape of the curves shows obvious differences between the operable and inoperable patients’ DSC curves. In the operable cases, the Tm_2_ and Tm_3_ transitions are dominant, compared with inoperable cases, Tm_3_, Tm_4_, and Tm_5,_ in which transitions were the dominant components. Several studies have confirmed a significant correlation between the thermal shift and modified structural stability of plasma proteins as disease-specific markers originating from cancer patients [13,14,18].

The contribution of the deconvolved components to the total calorimetric enthalpy can be seen in Figure 3. We observed that the enthalpy change shows a wide distribution in the range of transitions between 60–75 °C, in the case regarding patients afflicted with pancreatitis (Table 5). Surprisingly, the activation energy was increased in the case of pancreatitis and decreased in the case of tumors (Table 6).

### 3.3. Skin Cancer Studies

We intend to present the surgical treatment of skin cancer and melanoma malignum (MM) in the reevaluation of our former results [34,35,39]. We examined five healthy controls and fifteen ill Caucasian adult patients (twelve males and three females) prior to surgery. The Breslow parameter changed from 0.5 mm to 8.3 mm and the Clark level was II and IV. In the case of patients in which the Sentinel lymph node was positive (seven cases), adjuvant therapies (Interferon alpha-2b, Telecobalt) were implemented. The collection, preparation and storage of blood samples were made using the same protocol applied in our previous studies [34,35,39].

In Figure 4, the deconvolved DSC curves revealed that the main transition component of plasma from MM patients was shifted to a slightly higher temperature range. Although MM is a local malignant tumor, it does not cause any significant changes when compared with the control. However, in the case of local or distant metastases, the DSC curve is wider and more pronounced in the higher temperature ranges than in the control. However, the main component and the enthalpy change were slightly reduced due to the metastasis formation. In the case of melanoma, the main component of enthalpy change shows a narrow distribution in the range of Tm values between 60–63 °C (Table 7). As observed in the case of local and distant tumors, the enthalpy of blood components increased in the ranges of higher Tm values, between 64–72 °C (Table 7). The activation energy decreased in the case of the melanoma and increased in case of local tumors, however, there is no difference between the control and distant tumorous samples (Table 8).

## 4. Summary

As previously suggested, cancer formation can cause significant modification in the composition of blood plasma, which interacts directly/indirectly with all the cells in the human body and can be easily sampled [84]. The distribution pattern originating from a vast amount of plasma proteins between different categories varies as a monitoring signal of the individual’s health status. The screening of disease and diagnosis or in the monitoring of treatment highlighting changes of specific, or combined plasma proteins, serve as biomarkers. We can apply plasma protein concentrations and thermal parameters as good markers of genetical and lifestyle indicators in relation to diseases. Herein, we summarized that the changes are related to the DSC scans, including their deconvolved components [34,35,36,37,38,39,40,43,44,45,46,47,48,49].

Based on the current treatment principles, the stage of breast cancer is now determined by the size of the tumor and the number of metastatic axillary lymph nodes or the presence of distant metastases. In relationship to breast tumor size and lymph node status, the thermoanalytic results are aligned to the clinical progression of the disease. In the case of control patients, between four and seven separable components can be detected through the deconvolution of the curves. In reference to the literature data [14,55], when examining the healthy controls, the main component was albumin, while in the case of tumorous patients, the peaks of immunoglobulins proved dominant; thus, their thermal shift can be used as good markers of direct immunological response. Moreover, the higher temperature transitions appear significant in patients whose tumor diameter is larger than 10 mm, yet less than 50 mm. In the case of breast cancer patients, the calorimetric enthalpy decreases inversely with tumor size, identically with the tumor metastases towards the sentinel lymph nodes. Some similar observations have been made by several teams investigating DSC studies of plasma from patients with skin and gynecological tumors, gastric adenocarcinoma, colorectal carcinoma, central nervous system tumors and autoimmune diseases. Based on these observations, seemingly, each tumor or disease may have a unique fingerprint-like thermal curve. Furthermore, in the case of cervical carcinoma, they have also shown a correlation with the stage of the disease and the shape of the curves [11,12,13,14,17,18,19,20,21,22,50,55,57,59,60]. Interestingly, the activation energy is dependent upon the tumor size or the number of metastatic lymph nodes. It would appear that the more sensitive (quick denaturation, low activation energy) plasma proteins level increases with the appearance of novel cancer cells. In 2019, Faroongsarng and colleagues described how tumor invasion through unusual metabolism results in significant changes of albumin stability. In their study, blood was taken from patients afflicted with various stages of breast cancers and suggested that the heat transitions of albumin increased with the maturation of breast cancer, which can be related to the increasing amount of albumin binding peptides [85]. Furthermore, the determination of activation energy can be a good indicator for further research.

In the case of any chronic medical conditions, such as pancreatitis, the DSC scan shows a wide distribution of heat transitions which can be interpreted as a more variable immunological response than in the case of cancer formation. In the blood plasma of patients afflicted with chronic pancreatitis, it is possible that the plasma contains more proteins with low stability than when compared with healthy samples. Additionally, from the examination of heat transitions, the local inflammation reflects systemic inflammatory processes. Lastly, the slight decrease in enthalpy likely reflects the destabilization of globulins, which play a prominent role in inflammation. In the case of pancreatic tumors, new and higher denaturation temperatures appeared, and the enthalpy changed and as an indicator of the overall thermal stability of the whole system, showed a decreasing trend, particularly in inoperable cases. The activation energy increases with pancreatitis severity and drops due to cancer formation. Seemingly, the more sensitive plasma proteins’ level increases with the appearance of novel pancreatic cancer cells. Presumably, the thermal analysis of blood plasma can be an adequate method for the prognosis of early stages of pancreatic tumor.

In the case of MM cancer, the curves shifted to the higher denaturation temperatures, which can be interpreted as a higher level of stable immunoglobulins. These changes were also confirmed in the published literature, in which the average control curves were obtained from a study of 100 healthy individuals. The unique variations in the blood plasma in healthy and MM patients raised the possibility of a new diagnostic technology [10,11,12,13]. The values of activation energy increase with the progression of MM stages as a good indicator of a strong immunological response with less sensitive immunoglobulins. Among our measurements, the heat transitions in MM cases with distant metastases shifted to higher denaturation temperatures. There are no other DSC data measured in patients with distant metastatic MM. However, in cases of invasive cervical cancer, it has been described that the DSC scan shifted progressively towards higher denaturation temperatures compared to high-grade intraepithelial squamous lesions. Recent research applied by Spanish researchers using the DSC analysis of plasma from 63 MM patients found no false-negative results and noted the method was able to detect changes in disease status prior to clinical diagnosis [22].

In conclusion, the DSC curves obtained in different tumor stages showed a good correlation regarding the severity and spread of the malignant disease. The differences in the calculated activation energies with increased transition temperatures of plasma components provide strong evidence of a direct relationship between the appearance of cancer cells and the sensitive type of plasma proteins (Figure 5). Recent deconvolution analyses of the DSC curves have partially revealed the biological variations in support of the data. This requires further investigation, however, potentially ushers in a new aspect to modern diagnostics.

## Figures and Tables

**Figure 1 cancers-14-06147-f001:**
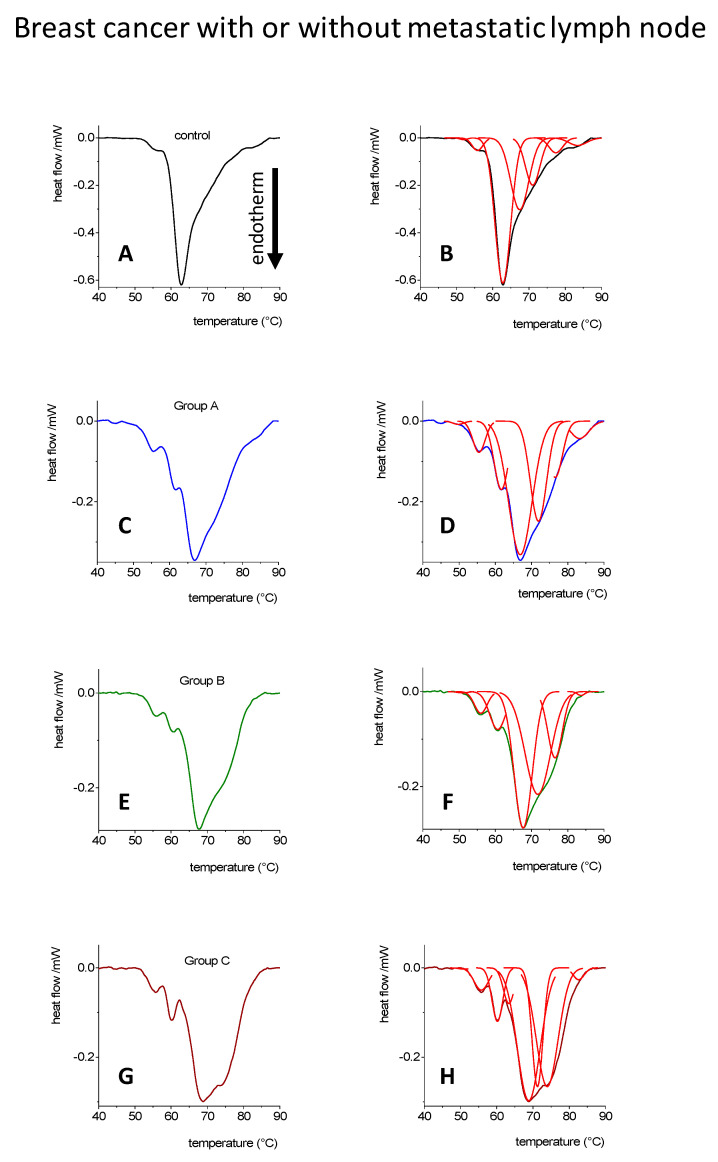
Denaturation scans of blood plasma (data were applied from our previous studies) [33,36,37] (**A**,**C**,**E**,**G**) and their deconvolution (**B**,**D**,**F**,**H**) from control patients (*n* = 10) and with breast tumor (*n* = 19), classified by different number of affected lymph nodes (see text).

**Figure 2 cancers-14-06147-f002:**
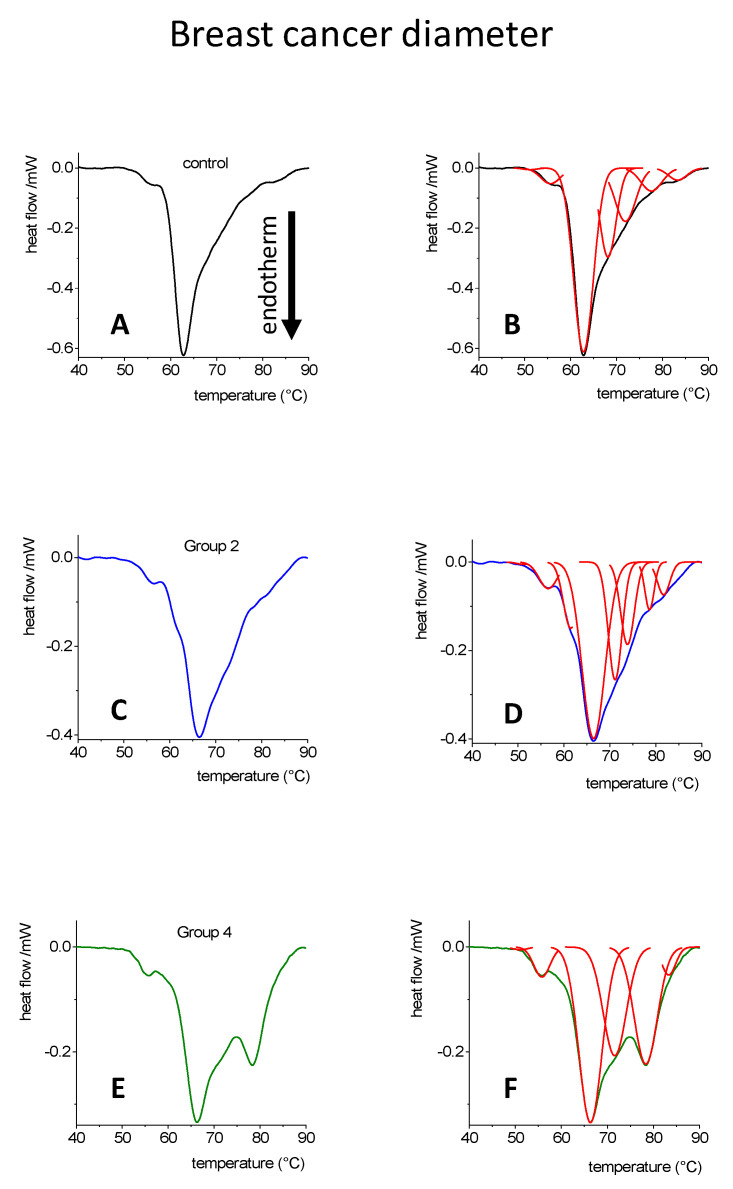
DSC curves [33,36,37] and their deconvolution in case of blood plasma from control individuals (**A**,**B**) (*n* = 10), and from breast cancer patients (*n* = 19), where the tumor diameter was between 11 and 20 mm in Group2 (**C**,**D**), and was doubled in size between 31 to 40 mm in Group4 (**E**,**F**).

**Figure 3 cancers-14-06147-f003:**
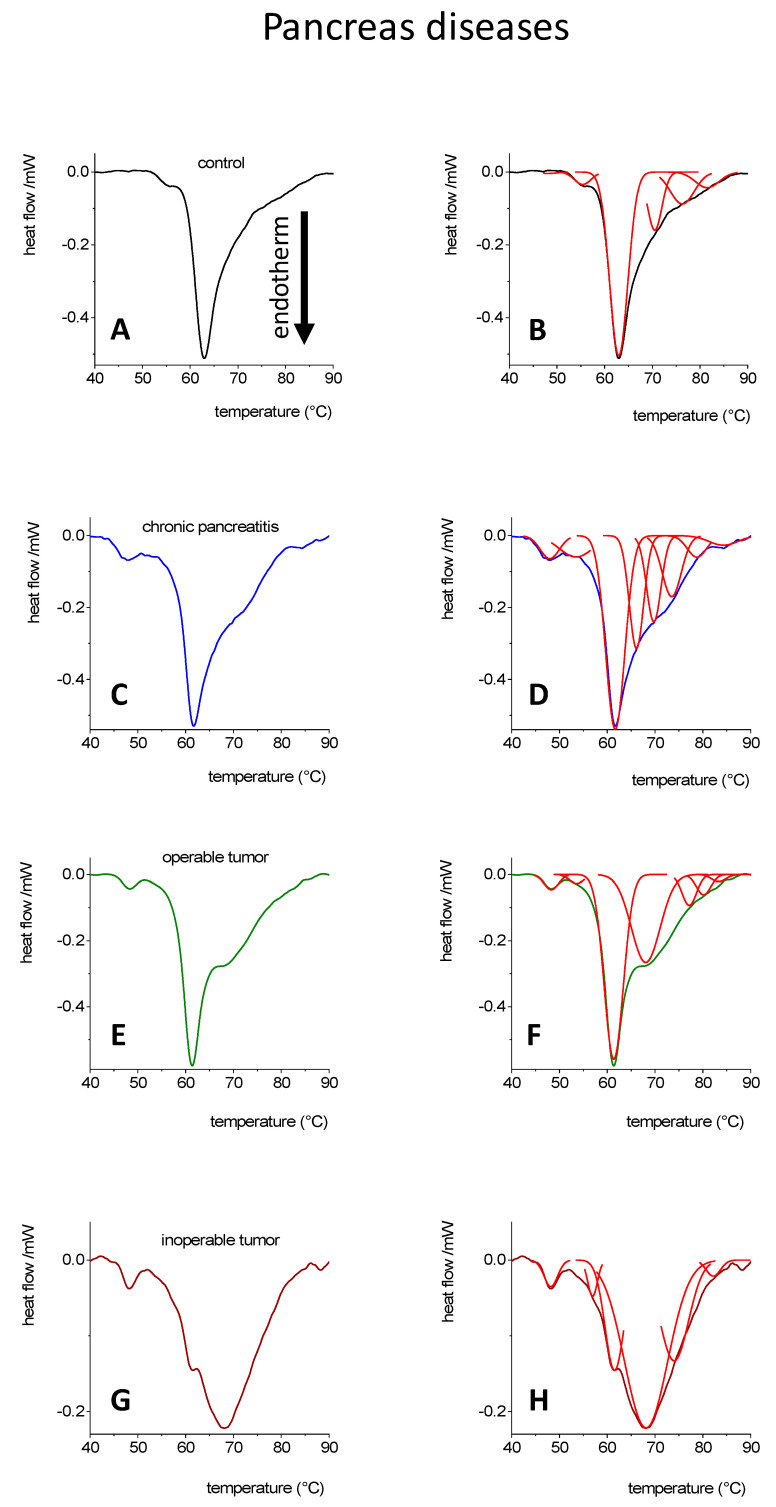
DSC curves [38] and their deconvolution in case of blood plasma from control (**A**,**B**, *n* = 5), chronic pancreatitis (**C**,**D**, *n* = 5), operable (**E**,**F**, *n* = 11) and inoperable pancreatic adenocarcinoma (**G**,**H**, *n* = 5).

**Figure 4 cancers-14-06147-f004:**
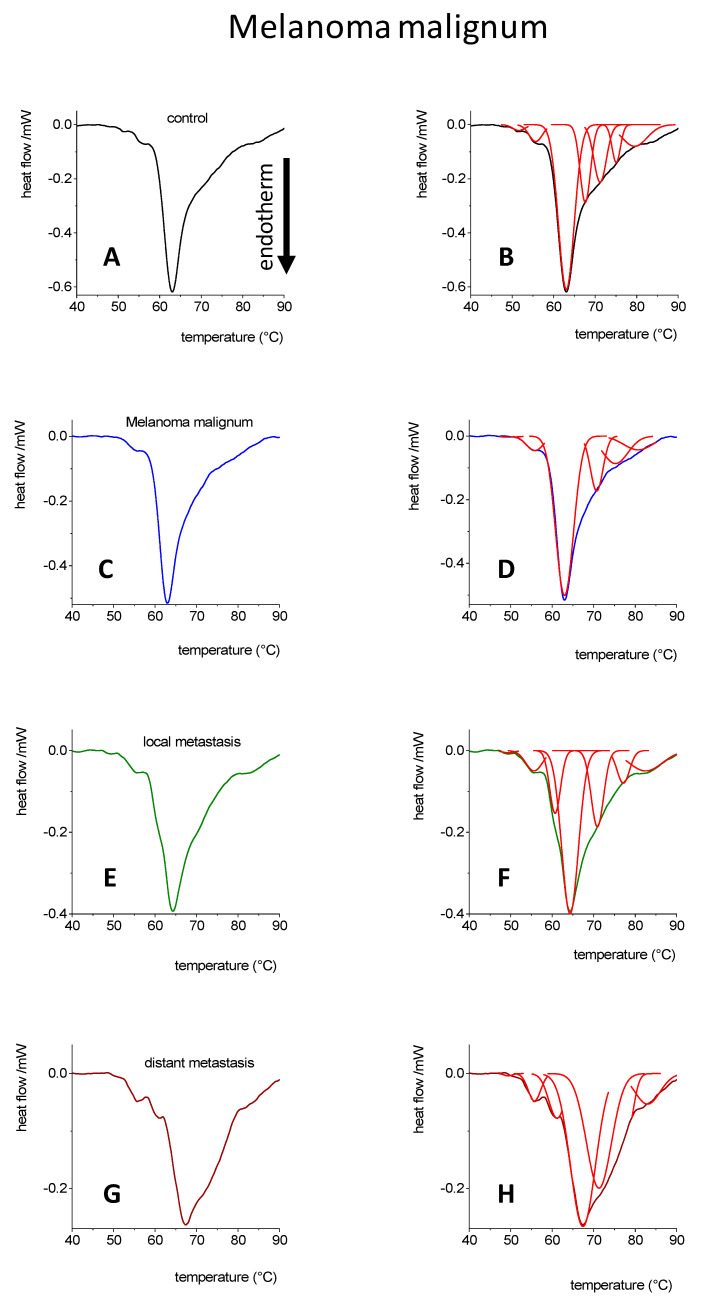
DSC curves [34,35] and their deconvolution in case of blood plasma from control (**A**,**B**), melanoma malignum (**C**,**D**), local metastasis (**E**,**F**) and distant metastasis (**G**,**H**). The number of healthy control patients was n = 5 and from patients afflicted with different stage of melanoma n = 15.

**Figure 5 cancers-14-06147-f005:**
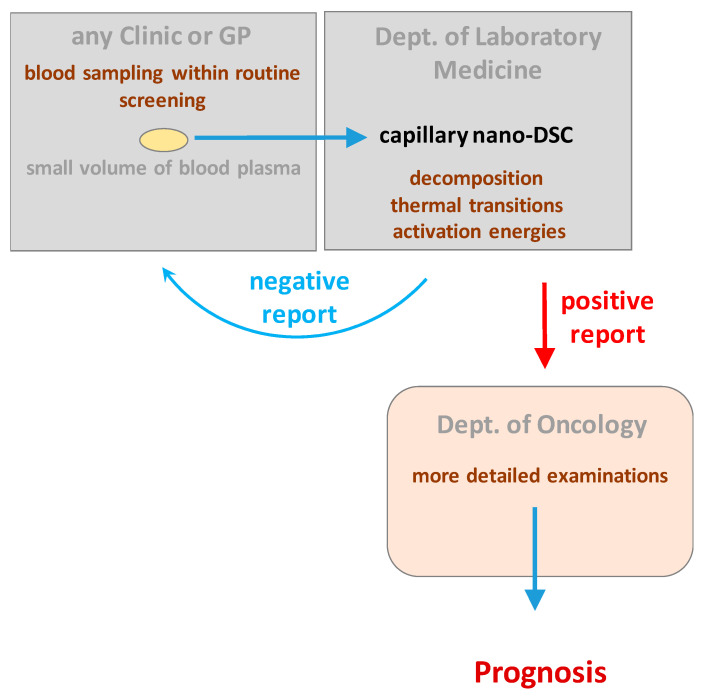
Summary figure to explain how DSC may be particularly relevant in clinical diagnosis.

**Table 1 cancers-14-06147-t001:** Contribution of deconvolved transitions from Figure 1 to the total calorimetric enthalpy, gray numbers infers the relative participation to the total change (based on instrument error, only those compounds are used which have a higher than 5% change, recently evaluated).

*Sample*	ΔH_cal_ (J/g)	T_m_ (°C) (%)	T_m_ (°C)(%)	T_m_ (°C) (%)	T_m_ (°C) (%)	T_m_ (°C) (%)	T_m_ (°C) (%)
*control*	1.41		62.8 *(64.4)*	68.1 *(14.9)*	71.9 *(11.7)*		77.6 *(5.1)*
*GroupA*	1.19	55.5 *(6.6)*	61.7 *(11.6)*	66.9 *(50.6)*	71.9 *(17.7)*		76.5 *(10.5)*
*GroupB*	0.88	56 *(5.1)*	60.7 *(7.1)*	67.7 *(52)*	71.7 *(23.5)*		76.5 *(11.7)*
*GroupC*	1.05	55.7 *(5)*	60.2 *(8.7)*	68.8 *(39.5)*	71.2 *(15.6)*	73.9 *(28.2)*	

**Table 2 cancers-14-06147-t002:** Contribution of activation energies (gray numbers) resulted by the analysis of deconvolved data from Figure 1 representing different temperature ranges of the peaks (ΔT). (The empty fields indicating missing data as the model fitting does not work properly with small peaks, and thus cannot result applicable values, recently evaluated).

*Sample*	E_a_ Total (kJ/mol)	E_a_ (kJ/mol) ΔT *(55–57)*	E_a_ (kJ/mol) ΔT *(60–63)*	E_a_ (kJ/mol) ΔT *(66–69)*	E_a_ (kJ/mol) ΔT *(70–72)*	E_a_ (kJ/mol) ΔT *(73–75)*	E_a_ (kJ/mol) ΔT *(76–79)*
*control*	* 3859.3 *		703.1	182.3	1845.7		1128.2
*GroupA*	* 3646.9 *		1162	877.4	1058.4		549.1
*GroupB*	* 2856.5 *		1025.1	932.9	447.2		451.3
*GroupC*	* 4007.2 *		1290.1	806.7	443.3	1467.1	

**Table 3 cancers-14-06147-t003:** Contribution of deconvolved transitions from Figure 2 to the total enthalpy change, gray numbers represent the relative participation to the total change (based on instrument error, only those compounds are used which have a higher than 5% change, recently evaluated).

*Sample*	ΔH_cal_ (J/g)	T_m_ (°C) (%)	T_m_ (°C) (%)	T_m_ (°C) (%)	T_m_ (°C) (%)	T_m_ (°C) (%)
*control*	1.41	62.8 *(64.4)*	68.1 *(14.9)*	71.9 *(11.7)*		77.6 *(5.1)*
*Group2*	1.26	61.5 *(6.1)*	66.4 *(54.3)*	71.1 *(12.1)*	73.8 (13)	78.6 *(5.4)*
*Group4*	1.17		66.3 *(52.9)*	71.6 *(12.1)*		78.3 *(27.5)*

**Table 4 cancers-14-06147-t004:** Contribution of activation energies (gray numbers) resulted by the analysis of deconvolved data from Figure 2 in the different temperature ranges of the peaks (ΔT). (The empty fields indicating missing data as the fitting does not work properly with small peaks, thus cannot result applicable values, recently evaluated).

*Sample*	E_a_ Total (kJ/mol)	E_a_ (kJ/mol) ΔT *(60–63)*	E_a_ (kJ/mol) ΔT *(66–69)*	E_a_ (kJ/mol) ΔT *(70–72)*	E_a_ (kJ/mol) ΔT *(73–75)*	E_a_ (kJ/mol) ΔT *(76–79)*
*control*	* 3175.3 *	951.4	674.2	206.3		1343.4
*Group2*	* 4336.7 *	993.9	975.3	1706.5	184.9	476.1
*Group4*	* 2686.9 *		888.1	864.2		934.5

**Table 5 cancers-14-06147-t005:** Contribution of deconvolved transitions from Figure 3 to the total enthalpy change, gray numbers represent the relative participation to the total change (based on instrument error, only those compounds are used which have a higher than 5% change, recently evaluated).

*Sample*	ΔH_cal_ (J/g)	T_m_ (°C) (%)	T_m_ (°C) (%)	T_m_ (°C) (%)	T_m_ (°C) (%)	T_m_ (°C) (%)	T_m_ (°C) (%)
*control*	1.12		62.5 *(73.4)*		70.5 *(14.1)*		76.3 *(7.2)*
*chronic*	1.54	48 *(5.4)*	61.7 *(51.1)*	66.1 *(14.6)*	69.7 *(11.9)*	73.5 *(9.6)*	
*operable*	1.47		61.4 *(61.4)*	68.31 *(27.8)*			
*inoperable*	0.85		61.5 *(14.3)*	68.1 *(61.4)*		74.7 *(13.7)*	

**Table 6 cancers-14-06147-t006:** Contribution of activation energies (gray numbers) resulted by the analysis of deconvolved data from Figure 3 regarding the different temperature ranges of the peaks (ΔT). (The empty fields indicating missing data as the fitting does not work properly with small peaks, thus cannot result applicable values, recently evaluated).

*Sample*	E_a_ Total (kJ/mol)	E_a_ (kJ/mol) ΔT *(55–57)*	E_a_ (kJ/mol) ΔT *(60–63)*	E_a_ (kJ/mol) ΔT *(66–69)*	E_a_ (kJ/mol) ΔT *(70–72)*	E_a_ (kJ/mol) ΔT *(73–75)*	E_a_ (kJ/mol) ΔT *(76–79)*
*control*	* 3745.8 *		970.4		1440.3		1335.1
*chronic*	* 5265.2 *	848.8	941.9	1530.7	579.9	1363.9	
*operable*	* 1613.3 *		927.6	685.7			
*inoperable*	* 2374.9 *		944	501.3		929.6	

**Table 7 cancers-14-06147-t007:** Contribution of deconvolved transitions from Figure 4 to the total enthalpy change, gray numbers represent the relative participation to the total change (based on instrument error, only those compounds are used which have a higher than 5% change, recently evaluated).

*Sample*	ΔH_cal_ (J/g)	T_m_ (°C) (%)	T_m_ (°C) (%)	T_m_ (°C) (%)	T_m_ (°C) (%)	T_m_ (°C) (%)	T_m_ (°C) (%)	T_m_ (°C) (%)	T_m_ (°C) (%)
*control*	1.51		63 *(56.8)*		67.6 *(13.4)*	71.2 *(10.5)*	75.1 *(7.2)*		
*melanoma*	1.14		63 *(73.6)*			70.7 *(11.3)*	75.3 *(8.2)*		
*local*	1.11		60.7 *(7.5)*	64.3 *(59)*		70.9 *(16.2)*		77.1 *(6.8)*	82.5 *(5.4)*
*distant*	0.92	55.7 *(5)*	61.2 *(6.4)*		67.4 *(46.6)*	71.3 *(27.7)*		78.1 *(8.1)*	83 *(5.8)*

**Table 8 cancers-14-06147-t008:** Contribution of activation energies (gray numbers) resulted by the analysis of deconvolved data from Figure 4 in the different temperature ranges of the peaks (ΔT). (The empty fields indicating missing data as the fitting does not work properly with small peaks, thus cannot result applicable values, recently evaluated).

*Sample*	E_a_ Total (kJ/mol)	E_a_ (kJ/mol) ΔT *(55–57)*	E_a_ (kJ/mol) ΔT *(60–63)*	E_a_ (kJ/mol) ΔT *(64–65)*	E_a_ (kJ/mol) ΔT *(66–69)*	E_a_ (kJ/mol) ΔT *(70–72)*	E_a_ (kJ/mol) ΔT *(73–75)*	E_a_ (kJ/mol) ΔT *(76–79)*	E_a_ (kJ/mol) ΔT *(80–84)*
*control*	* 5366.7 *		1320.9		2209.1	1492.9	206.4		
*melanoma*	* 3270.3 *		893.9			2105.2	271.2		
*local*	* 6595.3 *		1507.7	1133.5		1818.8		1475.9	658.4
*distant*	* 5108.1 *	1357.8	979.8		682.4	692.7		724.6	670.8

## Data Availability

The study reported all data.

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
