# Peer review of "Thermodynamic Sensitivity of Blood Plasma Components in Patients Afflicted with Skin, Breast and Pancreatic Forms of Cancer"

_cancers, 2022, doi:10.3390/cancers14246147_

Round 1
Reviewer 1 Report
The authors reviewed the potential usefulness of deconvolution analysis for serum DSC curves for diagnosis, staging, and size of several cancers based on the previous data. The review was well written and worthy of publication while there is just one minor comment.
1. In the legend of Figure 1 on page 5, the description of deconvolution should be corrected to (B, D, F, H).
Author Response
Summary: We thank the Editorial Board Members and Reviewers for their helpful and constructive comments. Additions or modifications made in direct responses are shown below. Changes made in response to those comments.
Reviewer Comments:
Answers to Reviewer 1.
- In the legend of Figure 1 on page 5, the description of deconvolution should be corrected to (B, D, F, H).
Thank you, for these important observations. We made the correction.
Reviewer 2 Report
There is undoubtedly an interest in using Differential Scanning Calorimetry as a diagnostic method. Several publications in the literature apply DSC to many clinical areas, especially cancer patient health status, diagnosis, and monitoring.
The presented article concerns the analysis of blood plasma samples denaturation profiles taken from breast, pancreatic, and skin cancer patients. The review is based mainly on evaluating the author's measurements (the previous and the newly performed ones) according to the newly evaluated parameters of the DSC curves. All the experiments have been carried out correctly with much field expertise and comply with the ethics guidelines. The presented approach, involving the deconvolution analysis of the DSC curve into peaks assigned to specific proteins groups denaturation and activation energy calculation, brings interesting information and allows authors to the conclusions with clinical relevance. The results discussion and conclusions are convincing and may be helpful for someone wanting to go deeper into this matter. The paper should be published after being thoroughly revised.
The reviewer remarks to the article are the following:
1. An article in the form of a review should cover a broader approach to selected three type of cancers. The introduction and partly the conclusion sections could be rewritten to make the review more informative for Cancers journal audience.
· "Introduction" part: it is suggested to focus mainly on recent publications (last 5 years), as expected by MDPI journals for review articles. It may turn out to be more valuable to the reader.
· According to the reviewer, the historical background is interesting at the beginning and provides a background for the beginnings of the application of the DSC method in the field of cancer diagnosis and health status monitoring, to which the special issue is devoted. However, the thematically discussed types of cancer can result in a structured and thoroughly discussed presentation of the state with a broad view of international publications.
· Chapter 4. is entitled "Conclusion." The authors could consider renaming this part as "Summary" - a more informative one that seems more suitable due to the review type of article. In a few cases in this part, the authors should deepen the discussion of their own results compared to the results of other teams. E.g., lines 378-381 – the authors mention that their research group was "(…) one of the first which demonstrated the diagnostic importance of DSC scans in cases of MM(…)". The other results should be after that presented.
2. Perhaps it would be better if the number of samples represented by each DSC curve in Fig. 1, 2, and 3 have been included. Moreover, references in the legend would clarify whether the denaturation curve is given from published articles or new measurements.
3. In Fig. 1H in the pdf version of the manuscript, some parts of the DSC curves are invisible. Please check what causes this error (image is fully visible in non-published files).
4. The authors could provide a brief description to explain what information the activation energy, a thermodynamic parameter, based on which the authors assess the impact of cancer on the stability and kinetics of denaturation of plasma components, carries here. (line 185 ).
5. It would be good to include a summary figure at the end of the article explaining how DSC may be particularly relevant in clinical diagnosis and treatment monitoring of those particular diseases.
Author Response
Summary: We thank the Editorial Board Members and Reviewers for their helpful and constructive comments. Additions or modifications made in direct responses are shown below. Changes made in response to those comments.
Answers to Reviewer 2.
The reviewer remarks to the article are the following:
An article in the form of a review should cover a broader approach to selected three type of cancers. The introduction and partly the conclusion sections could be rewritten to make the review more informative for Cancers journal audience. "Introduction" part: it is suggested to focus mainly on recent publications (last 5 years), as expected by MDPI journals for review articles. It may turn out to be more valuable to the reader.
According to the reviewer, the historical background is interesting at the beginning and provides a background for the beginnings of the application of the DSC method in the field of cancer diagnosis and health status monitoring, to which the special issue is devoted. However, the thematically discussed types of cancer can result in a structured and thoroughly discussed presentation of the state with a broad view of international publications.
Thank you for this suggestion. We did an effort to modify the rest of the text in the Part of Introduction and the new Summary as well. Nevertheless, we listed near all important publications that have been published on the subject. This field is not so intensively researched and published that the last 5 years would give a sufficient overview of the subject.
Chapter 4. is entitled "Conclusion." The authors could consider renaming this part as "Summary" - a more informative one that seems more suitable due to the review type of article. In a few cases in this part, the authors should deepen the discussion of their own results compared to the results of other teams. E.g., lines 378-381 – the authors mention that their research group was "(…) one of the first which demonstrated the diagnostic importance of DSC scans in cases of MM(…)". The other results should be after that presented.
Thanks for the Reviewer for these important recommendations. Already, we did all modifications and hopefully improved it.
Perhaps it would be better if the number of samples represented by each DSC curve in Fig. 1, 2, and 3 have been included. Moreover, references in the legend would clarify whether the denaturation curve is given from published articles or new measurements.
In Fig. 1H in the pdf version of the manuscript, some parts of the DSC curves are invisible. Please check what causes this error (image is fully visible in non-published files).
The authors could provide a brief description to explain what information the activation energy, a thermodynamic parameter, based on which the authors assess the impact of cancer on the stability and kinetics of denaturation of plasma components, carries here. (line 185).
It would be good to include a summary figure at the end of the article explaining how DSC may be particularly relevant in clinical diagnosis and treatment monitoring of those particular diseases.
Thank you for these important suggestions, we corrected and tried to improve the whole manuscript.
Reviewer 3 Report
This is a review article in which the authors have summarized their previous DSC research on blood plasma isolated from patients with three different types of cancer, breast cancer, melanoma and pancreatic cancer. They have identified certain, clearly discernible distinctions between control thermograms typical of healthy individuals and the thermograms of plasma isolated from patients at various stages of the tumor development. They have applied thermogram deconvolution into sets of Gaussian components in an attempt to more clearly distinguish and quantify cancer-induced changes in the plasma heat capacity profiles. This is a useful and informative approach, although the Gaussian peaks obtained by deconvolution in many cases cannot be assigned a clear physical meaning. For example, they might well be due to asymmetric s denaturational transitions that cannot be represented with a single Gaussian peak. In the course of this work the authors have reached some interesting conclusions. They point out that a correlation appears to exist between the degree of deviation of the thermogram shape from the control thermogram and cancer severity and degree of tumor development. It is also of interest that the three studied types of cancer appear to be accompanied by different effects on the thermogram shapes.
Overall, the manuscript is written on a good professional level. The authors are obviously experts in this field and they have been able to present their experimental observations and ensuing conclusions in a clear and transparent manner.
p. 3, lines116-118 The explanation that something has been omitted is not clear and needs to be rewritten.
Related to the previous comment, it would be helpful if the authors could add detailed information on the deconvolution accuracy and the uniqueness of the number of Gaussian peaks used to represent a given region of the thermogram.
Author Response
Summary: We thank the Editorial Board Members and Reviewers for their helpful and constructive comments. Additions or modifications made in direct responses are shown below. Changes made in response to those comments.
Reviewer Comments:
Answers to Reviewer 3.
Overall, the manuscript is written on a good professional level. The authors are obviously experts in this field and they have been able to present their experimental observations and ensuing conclusions in a clear and transparent manner.
- 3, lines116-118 The explanation that something has been omitted is not clear and needs to be rewritten.
Related to the previous comment, it would be helpful if the authors could add detailed information on the deconvolution accuracy and the uniqueness of the number of Gaussian peaks used to represent a given region of the thermogram.
Thank you for this important observations, we corrected them and gave a more accurate interpretation in the part of Materials and Methods.
Round 2
Reviewer 2 Report
The manuscript has been cerrected according to the suggestions. The review is seen very interesting for Cancers journal audience.